# Conditional Deontics over Terminals: A Mildly Context-Sensitive Formal Grammar for Constrained Decoding

## Abstract

Constrained decoding researchers have recently thought to extend context-free constrainers to context-sensitive constraints such as matching selected columns to tables in text-to-SQL. This is challenging, because context-sensitivity is achieved by cross referencing different subtrees in the syntactic structure, which is only partially available during LLM generation. Recent frameworks such as IterGen gain some context sensitivity but with drawbacks including the need for expensive backtracking or speculative lookahead of LLMs, overly permissive semantics risking over-constraint, and complications in interfacing with works on improving running speed such as XGrammar. To address these concerns, we propose a new mildly-context-sensitive formal grammar called Conditional Deontics over Terminals (CDoT). Its incremental parser has $O(n^3)$ time complexity each step and allows for GPU acceleration, compared to the $O(n^8)$ of mildly context-sensitive tree-adjoining grammars and the $O(n^2)$ of context-free grammars. This new formal grammar is strong enough to implement the unit propagation algorithm, which we employ to assist an LLM in solving the "Knight and Knave" logical puzzle, achieving substantially improved performance at reduced output token budget. We also evaluate on the traditional constrained decoding task of text-to-SQL.

## 1 Introduction

Constrained decoding allows programmatic control of inference-time LLM behavior without modifying model weights or engineering prompts. Prior work has successfully employed it to enforce programming language specifications, molecular structures, and privacy requirements (Ugare et al., 2025; Loula et al., 2025; Scholak et al., 2021; Poesia et al., 2022).

As the technique matures, works have emerged that improve various aspects of it. One line of inquiry pushes for constraints of higher complexity class via speculative look-ahead or LLM backtracking (Ugare et al., 2025; Loula et al., 2025). This is not only computationally expensive—requiring user code to access and modify parser states—but also complicates integration with runtime optimization methods such as XGrammar (Dong et al., 2025). The semantics allowed by such frameworks can also sometimes be overly permissive and risk hazardous over-constraints that are difficult to detect.

To address these concerns, we design a new formal grammar for constrained-decoding, with the aim of shielding users from the complexities of constrainer internals while providing researchers with a stable target interface. Our contributions are:

- We design a new formal grammar that offers context-sensitivity in practical use cases and can be efficiently parsed incrementally at $O(n^3)$ each step.
- Our formal grammar allows invocation of Python functions for added flexibility, while eliminating the need for user code to access and modify parser state.
- We implement a GPU-accelerated parser for our grammar by extending an incremental Valiant recognizer.

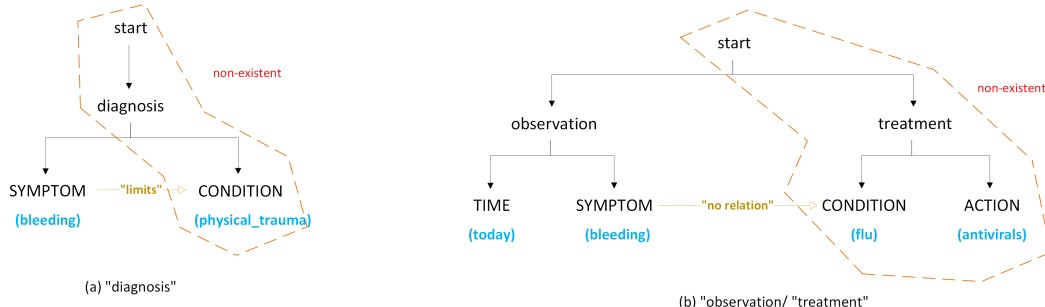

(a) "diagnosis"

(b) "observation/ "treatment"

Figure 1: Example: a generated symptom should constrain the associated condition that is generated. But when `CONDITION` is being generated its ancestor structure (dashed lines) has not yet been determined due to the nature of incremental parsing. Knowing the possible ancestor structures is important, as the symptom-condition constraint should fire on the left tree but not the right.

## 2 RELATED WORK

**Incrementally Parsable Mildly Context-sensitive Grammars:** Constrained decoding relies on incremental parsers capable of maintaining the valid-prefix property. Despite a diverse selection of mildly context-sensitive grammars (Joshi & Schabes, 1997; Vijay-Shanker & Weir, 1993; Okhotin, 2001; Mrykhin & Okhotin, 2023; Barash & Okhotin, 2014), such grammars are often not designed with the goal of efficient constrained-decoding in mind. For example, Tree-Adjoining Grammars have an $O(n^8)$ step time-complexity (Schabes, 1991). The recently published grammar with one-sided context (Barash & Okhotin, 2014) within the conjunctive grammar family overlaps with the goal of constrained decoding, but is currently a theoretical construct applied only to toy grammars.

**Constrained Decoding with Context Sensitivity:** Context-sensitivity may reference across sub-trees, as in fig. 1 where the `SYMPTOM` subtree must be observed to determine which `CONDITION`s are allowable. In the case of constrained decoding, an incremental process, the ancestor nodes are unavailable at the time a constraint must be applied. Earlier work such as PiCARD (Scholak et al., 2021) ignored ancestor structure, selecting preceding terminals directly and risking constraint mis-fires as would be the case in the right side of fig. 1. IterGen (Ugare et al., 2025) and GenLM Control (Loula et al., 2025) allowed ancestor structures to be considered by invoked expensive backtracking and/or speculative look-ahead, where ancestor structures can emerge before constraints are applied (Ugare et al., 2025; Loula et al., 2025). Melcer et al. (2024) implements specialized context-sensitive constraints for some lexing features in programming languages.

## 3 FORMALISM FOR CONDITIONAL DEONTICS OVER TERMINALS (CDoT)

A *context-free grammar (CFG)* is a 4-tuple $G = (V, \Sigma, R, S)$ where

- $V$ is a finite set of *nonterminal symbols* (also called variables), which represent syntactic categories that can be expanded,
- $\Sigma$ is a finite set of *terminal symbols*, which are the basic alphabet symbols that appear in the strings of the language, with $V \cap \Sigma = \varnothing$,
- $R \subseteq V \times (V \cup \Sigma)^*$ is a finite set of *production rules* of the form $A \to \alpha$, where $A \in V$ (a nonterminal) and $\alpha \in (V \cup \Sigma)^*$ (a string of terminals and/or nonterminals),
- $S \in V$ is the designated *start symbol*.

The language generated by $G$ is defined as $L(G) = \{ w \in \Sigma^* \mid S \Rightarrow^* w \}$, where $\Rightarrow^*$ denotes the reflexive, transitive closure of the single-step derivation relation $\Rightarrow$ induced by $R$. A language $L \subseteq \Sigma^*$ is called *context-free* if there exists a CFG $G$ such that $L = L(G)$.

We extend CFGs with *deontic rules*, which constrain not only the structure of derivations, but also the *obligations* and *permissions* of terminals. A *Conditional Deontics over Terminals (CDoT)* grammar is a pair $G^+ = (G, D)$ where

- $G = (V, \Sigma, R, S)$ is a context-free grammar (possibly with regular-expression terminals),
- $D$ is a finite set of deontic rules, which constrain terminal instances generated by $G$.

Each deontic rule has an *anchor*, identifying the scope of the rule, and a *deontic function*, which maps terminal instances to a pair $(\Box C, \Diamond P)$, where $\Box C$ is a set of regular expressions (necessary conditions), and $\Diamond P$ is a set of regular expressions (permitted conditions).

**Non-conditional deontic rules.** A non-conditional rule has the form

$$X^{consequent} \ll t^{conseqent},$$

with $X^{conseqent} \in V$ and $t^{conseqent} \in \Sigma$. When the deontic function of a non-conditional deontic rule returns $(\Box C, \Diamond P)$, the semantics are that all terminals descended from $X$ must *collectively* satisfy every $r \in \Box C$, and each individual terminal must match at least one $r \in \Diamond P$.

**Example (Balanced Meals).** Given the grammar:

$$meal \rightarrow food \; beverage$$
$$food \rightarrow \texttt{DISH} \; food \; | \; \texttt{DISH}$$
$$beverage \rightarrow \texttt{BEVERAGE} \; beverage \; | \; \texttt{BEVERAGE}.$$

The following deontic rule enforces that every meal includes grains, protein, and vegetables ($\Box$), while also allowing optional extras like `soup` or `steamed_egg` ($\Diamond$).

```
anchor: food ≪ DISH
def enforce_meal():
    return (□{rice|bread, chicken|fish,
            salad|roasted_vegetable},
          ◊{rice, bread, chicken, fish,
            salad, roasted_vegetable, steamed_egg, soup})
```

**Conditional deontic rules.** A conditional rule has the form

$$t^{antecedent} \gg X^{antecedent} B \ldots D > X^{parent} < X^{consequent} \ll t^{consequent},$$

where $t^{antecedent}, t^{consequent} \in \Sigma$, and $B, D, X^{antecedent}, X^{consequent} \in V$ are variables appearing in a parent production $X^{parent} \rightarrow \ldots X^{antecedent} B \ldots D X^{consequent} \ldots$. The deontic function of a non-conditional deontic rule takes an antecedent terminal instance $t^{antecedent}$ as input, and uses it to determine and return constraints $(\Box C, \Diamond P)$, on the consequent terminal $t^{consequent}$.

**Example (Food–Drink Pairings).** Given the same grammar as the previous example, the following deontic rule enforces that when a food and a beverage are paired within a meal, steaks require red wine, fish requires white wine, and spicy noodles require soda.

```
anchor: DISH ≫ food > meal < beverage ≪ BEVERAGE.
def match_drink_to_food(terminal_instance):
    match terminal_instance.text:
        case 'steak':
            return (□{'red_wine'}, ◊{})
        case 'fish':
            return (□{'white_wine'}, ◊{})
        case 'spicy_noodle':
            return (□{'soda'}, ◊{})
```

Notice that, while a single antecedent variable *food* can cover multiple antecedent `DISH` variables, each invocation of the deontic function $match\_drink\_to\_food$ sees only a single instance of `DISH`. This prevents repetitive calling of the deontic function over different combinations. At the same time, the conjunctive nature of $\Box$ and disjunctive nature of $\Diamond$ will still naturally combine so that, if both fish and steak are ordered, both red and white wine are requested.

## 4 PARSER

CYK-like parsers use **TABLE**$[row, col, nt]$ to denote $nt \Rightarrow *t_{row} \ldots t_{row+col}$. To also track deontic rules we introduce a new state type **LF**$[row, col, cnf]$, which indicates a property of a CNF context-free rule with its left foot covering $t_{row} \ldots t_{row+col}$, and the CNF rule itself covering $t_{row}$ to any possible future terminal.

Our parser consists close to 20 slightly modified incremental valiant recognizers computing different properties. For simplicity, we focus on **LF**$^{ALIVE}$, **LF**$^{REACHABLE\uparrow}$, **LF**$^{REACHABLE\downarrow}$.

**LF**$^{ALIVE}$ tracks left feet whose has $\Diamond$ has not yet being violated. All $\Box$ underneath the left foot is satisfied, although the left foot itself might not. By tracking invalidated $\Diamond$ deontics, we prevent co-mingling of deontics from incompatible parse trees. We compute this attribute when a new terminal $t_n$ is being committed.

**LF**$^{ALIVE}[row, col, cnf] = True$ if and only if any of

- it is deontic that permits the new terminal.
- it is a descendant or ancestor of a deontic that permits the new terminal.
- reachable from both top and bottom without going pass a CNF rules associated with deontic rule has $type(t_n)$ as consequent. (unconstrained version of $t_n$)
- is a newly formed left foot, cannot be exceeded yet

**LF**$^{REACHABLE\uparrow}$, **LF**$^{REACHABLE\downarrow}$ is computed when probing for prospective next terminal $t_n$. A left foot need to be both $REACHABLE \uparrow$ and $REACHABLE \downarrow$ for it to be in one of the completable parse tree of $t_0 \ldots t_n$. Both attributes can be computed by logging successful triggering of CNF rule during execution of incremental Valiant recognizer.

**Permitted deontics**  are actualized by only allowing deontics of antecedent terminals beneath deontic left feet that are $ALIVE$, $REACHABLE \uparrow$ and $REACHABLE \downarrow$ at the same time, conditioned on matching consequent terminal type to the prospective terminal.

**Necessary deontics enforcement**  simply happens at commit-time by preventing underlying CNF rules from finishing until the right foot is grown enough to satisfy all required regular expressions.

We further illustrate the different between a CDoT parser states and ordinary CYK parser state in figue 2.

## 5 EXPERIMENTS AND RESULTS

### 5.1 KNIGHT AND KNAVE

Knight and Knave is a logic puzzle where each character is either a knight (always tells the truth) or a knave (always lies). The task is to infer each character's type from their statements. For example, if Alice says "Alice and Bob have the same identity", and Bob says "Alice is lying", we must deduce that Alice is a knave and Bob is a knight. Xie et al. (2024) provides a readily usable dataset containing both textual and logical form of puzzles with 2-8 characters.

#### 5.1.1 PREPARATION: CONJUNCTIVE NORMAL FORM AND UNIT PROP

**Conjunctive Normal Form**  is a conjunction of *clauses*, each clause being a disjunction of *literals* (a variable or its negation). For example, $(x_1 \lor \neg x_3 \lor x_4) \land (\neg x_1 \lor x_3) \land (x_1 \lor x_5)$. An arbitrary boolean expression can be converted into equalsatisfiable conjunctive normal form efficiently via the Tseytin transformation (Tseitin, 1983) with a caveat of adding $O(n)$ auxiliary variables.

**Unit Propagation**  is an algorithm that deduces assignments based on committed assignments of other literals. It exploits the fact that each clause in conjunctive normal form must have at least one literal to be true. Therefore, if all but one literal in a clause are false, the last remaining literal must be true. In the example above, if we assume $x_1$ is true, then $x_3$ must also be true, because it is the last remaining option to keep the second clause satisfied.

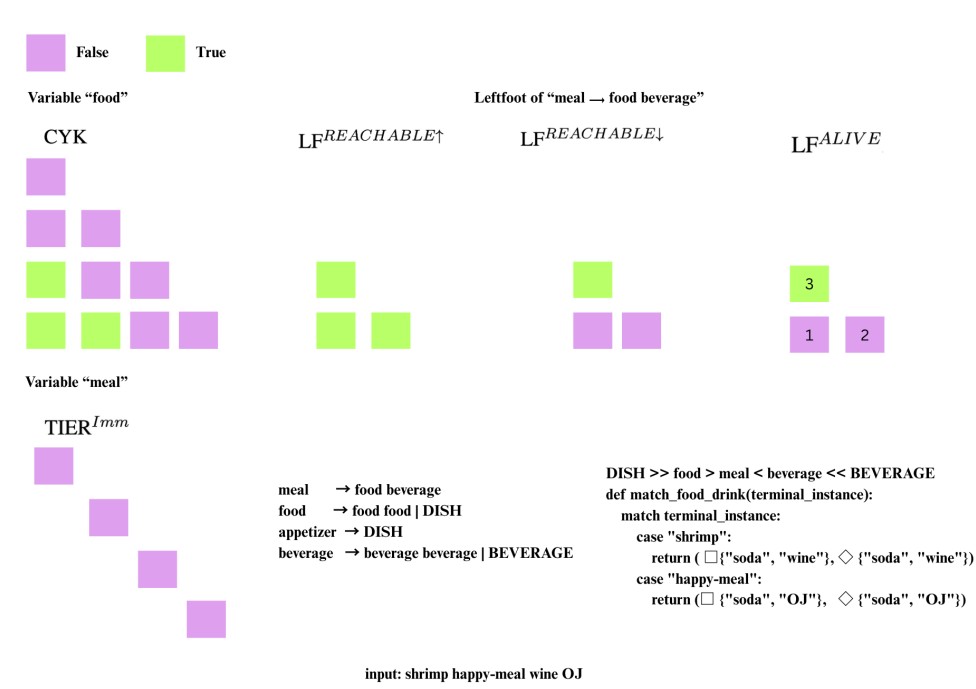

Figure 2: A CYK parser (context-free) and partial states of a CDoT parser after consuming input sequence "shirmp happy-meal wine OJ". There is no $Box$ guarding *food*, therefore $\mathbf{LF}^{REACHABLE\uparrow}$ for *meal → food beverage* behaves the same as CYK states for *food*. Position 3 in $\mathbf{LF}^{ALIVE}$ is true because it covers both "shrimp" and "happy-meal" to permit both "wine" and "OJ". Both position 1 and 2 are cannot permit both drinks by themselves. Although the left variable of both positions are in the same tree as 3, the left feet they represent are not. Therefore 1 and 2 cannot "inherent" aliveness by reachability. $\mathbf{TIER}^{IMM}$ does not form any *meal* instances because $\Box$ are not yet satisfied.

### 5.1.2 CONSTRAINER DESIGN

Figure 3 shows the context free grammar we employ for the output of the LLM on the Knight and Knave problem. It ensures that the model outputs a sequence of guesses. Each guess assigns roles to characters and booleans value to auxiliary variables. The grammar alone will not ensure the desired behavior, so we add the following deontic rules:

- The deontic rule for ensuring that *sufficient_assignment_seq* is recognized iff an *assignment_seq* commits to at least one literal from each clause is:

```
anchor: sufficient_assignment_seq << ASSIGNMENT
% attr: hidden\_permitted
def correct_guess():
    return (□{ "x_i|...|x_j" for x_i ∨ ··· ∨ x_j in input_clauses}, ◇{})
```

- The deontic rule for constraining *contradictory_assignment_seq* is defined similarly.
- The deontic rule for ensuring that a REPEATED_ASSIGNMENT is recognized iff it replicates a previous ASSIGNMENT within this guess is defined as:

```
anchor: ASSIGNMENT >> assignment_seq_semi_1 > repetition_assignment_seq <
        repeating_assignment << REPEATED_ASSIGNMENT
def repetition(terminal_instance):
    return (□{}, ◇{terminal_instance})
```

$$
\begin{aligned}
\textit{start} &\rightarrow \textit{bad\_guess} * \textit{good\_guess} \\
\textit{good\_guess} &\rightarrow \textit{sufficient\_assignment\_seq semi\_3}\ \text{GOOD\_GUESS\_TAIL} \\
\textit{sufficient\_assignment\_seq} &\rightarrow \textit{assignment\_seq} \\
\textit{bad\_guess} &\rightarrow \textit{contradictory\_assignment\_seq semi\_3}\ \text{CONTRADICTORY\_GUESS\_TAIL} \\
&\mid \textit{repetition\_assignment\_seq semi\_3}\ \text{REPETITION\_GUESS\_TAIL} \\
\textit{contradictory\_assignment\_seq} &\rightarrow \textit{assignment\_seq} \\
\textit{repetition\_assignment\_seq} &\rightarrow \textit{assignment\_seq\_semi\_1 repeating\_assignment} \\
\textit{assignment\_seq\_semi\_1} &\rightarrow \textit{assignment\_seq semi\_1} \\
\textit{assignment\_seq} &\rightarrow \textit{assignment\_seq semi\_1 assignment} \\
&\mid \text{GUESS\_HEAD}\ \textit{assignment} \\
&\mid \textit{unit\_prop} \\
\textit{assignment} &\rightarrow \text{ASSIGNMENT} \\
\text{ASSIGNMENT} &\rightarrow (\text{NAME "->" KK}) \\
&\mid \text{AUX\_VAR} \\
\textit{repeating\_assignment} &\rightarrow \text{REPEATING\_ASSIGNMENT} \\
\text{REPEATING\_ASSIGNMENT}^2 &\rightarrow \text{ASSIGNMENT} \\
\textit{semi\_3} &\rightarrow \text{SEMI\_3 "\textbackslash n"} \\
\textit{semi\_2} &\rightarrow \text{SEMI\_2 "\textbackslash n"} \\
\textit{semi\_1} &\rightarrow \text{SEMI\_1 "\textbackslash n"} \\
\text{SEMI\_3}^3 &\rightarrow \text{";"} \\
\text{SEMI\_2}^2 &\rightarrow \text{";"} \\
\text{SEMI\_1}^1 &\rightarrow \text{";"} \\
\text{GUESS\_HEAD} &\rightarrow \text{"Here is a different and improved guess: \{\{\textbackslash n"} \\
\text{GOOD\_GUESS\_TAIL} &\rightarrow \text{"\}\} <<DONE>>"} \\
\text{CONTRADICTORY\_GUESS\_TAIL} &\rightarrow \text{"\}\} contradiction\textbackslash n\textbackslash n"} \\
\text{REPETITION\_GUESS\_TAIL} &\rightarrow \text{"\}\} Bad guess\textbackslash n\textbackslash n"} \\
\text{NAME} &\rightarrow \text{/(Jacob|Noah|Michael|Liam|Ella|...|Ava)/} \\
\text{KK} &\rightarrow \text{"knight" | "knave"}
\end{aligned}
$$

Figure 3: Context free grammar for LLM generations in the Knight and Knave problem.

- Deontic rules for implementing the unit propogation algorithm are defined for each $x_k$ in the input clauses:
  - Add context-free rule "*unit_prop* → *x_k_condition semi_2 x_k_assignment*"
  - Add deontic rule

    ```
    anchor: x_k_assignment << ASSIGNMENT
    % attr: self\_sufficient\_necessary
    def x_k_condition():
        return (□{"x_k"}, ◇{})
    ```

  - For each input clause $c$ that contains $x_k$, add deontic rule

    ```
    anchor: x_k_condition << ASSIGNMENT
    % attr: hidden\_permitted
    def x_k_assignment():
        return (□{"NEGATE_LITERAL(x_m)" for x_m in c if m ≠ k},
                ◇{"x_m" if x_m exists in any clause and m ≠ k})
    ```

These deontic rules result in the real-time triggering of unit propagation and the real-time detection of good and bad guesses. As the LLM commits to literals, unit propagation will trigger immediately upon conditions being met, and will constrain generation to only entailed literals. Bad guesses will be detected immediately upon their first contradiction and generation will be forced to start a new guess. Good guesses will be detected immediately once the formula is solved and generation will be immediately terminated.

## 5.2 RESULTS

We set LLMs to solve puzzles in a 0-shot setting. Models are provided with both the textual and logical forms of the puzzle in the prompt. The results are shown in table 1. When the output is limited to 512 tokens, constrained models achieve about a 50-point higher solve rates than non-

| Model | Constrained | | Unconstrained | |
|---|---|---|---|---|
| | *Thinking* | *Non-thinking* | *Thinking* | |
| | *512* | *512* | *512* | *2048* |
| Llama-3.2-1B-Instruct | 0.52 | 0.04 | 0.04 | OOM |
| Llama-3.2-3B-Instruct | 0.65 | 0.13 | 0.13 | OOM |
| Qwen/Qwen3-0.6B | 0.61 | 0.07 | 0.07 | 0.36 |
| Qwen/Qwen3-1.7B | 0.69 | 0.03 | 0.03 | 0.55 |

Table 1: Knight and Knave test: solve rate of constrained vs. unconstrained models with 512 or 2048 token limits. Llama-3 models do not have a thinking switch, but do exhibit chain of thought-like behavior. Xie et al. (2024) reported 0.14 for Llama-3-8B-instruct in the same setting.

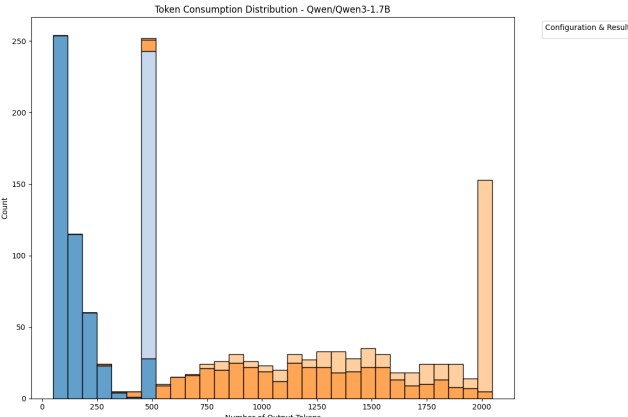

Figure 4: Knight and Knave test: token efficiency of successes (dark) vs failures (light) of constrained (blue) vs. unconstrained (orange) models. Constrained models were cut off at 512 tokens, unconstrained at 2048 tokens.

constrained models, and even outperform non-constrained models that are allowed to generate 4 times more tokens.

We also plot the number of output tokens for successful (dark) and failed (light) LLM responses across constrained (blue) and unconstrained (orange) Qwen3-1.7B, shown in fig. 4. The constrained model shows a skewed distribution, where a larger portion of puzzles are solved with lesser tokens, and failures don't show up until around 512 tokens. (Qualitatively, we observe that at this point, the constrained model gets stuck in loops.) The non-constrained model forms a flat bell curve distribution, and requires many more output tokens before starting to solve puzzles.

We also observe that the constrained model can often solve puzzle on the first attempt (fig. 5). This indicates that the constrainer and LLM are organically collaborating. If the constrainer alone was driving success, we would expect the distribution of guesses to be closer to random uniform, and it would be rare to solve the puzzle on the first guess. If the LLM alone was driving success, we would expect to see the non-constrained model having a similar number of solutions on the first attempt.

When we break down the solve rate by the size of the puzzle in fig. 6, we observe that the constrained models scale much better and still achieve solve rate of around 0.50, in contrast to under 0.20 for non-constrained models.

## 5.3 TEXT-TO-SQL

Spider (Yu et al., 2019) is a text-to-sql task, the model is provided with a database schema and a natural text question, and it is expected to answer the question with an generated SQL query.

We finetune an LLM with the constrainer on during both finetuning and inferencing. For the constrainer, we use a grammar extracted from the SQLite source code as the base context-free grammar, then add constraints that allow only identifiers that come from databases containing the columns in

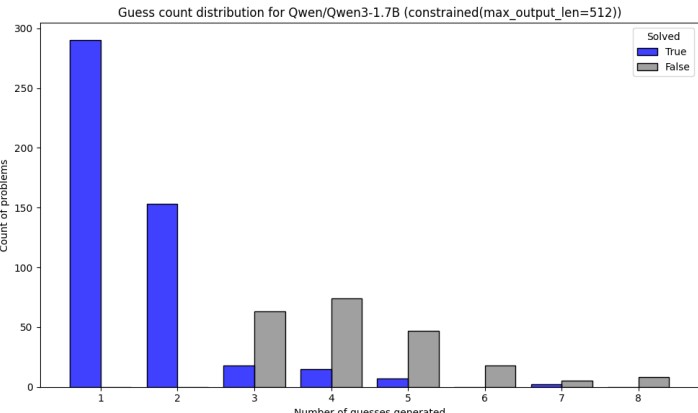

Figure 5: Knight and Knave test: constrained models solve many problems in the first guess, while unconstrained models require more guesses.

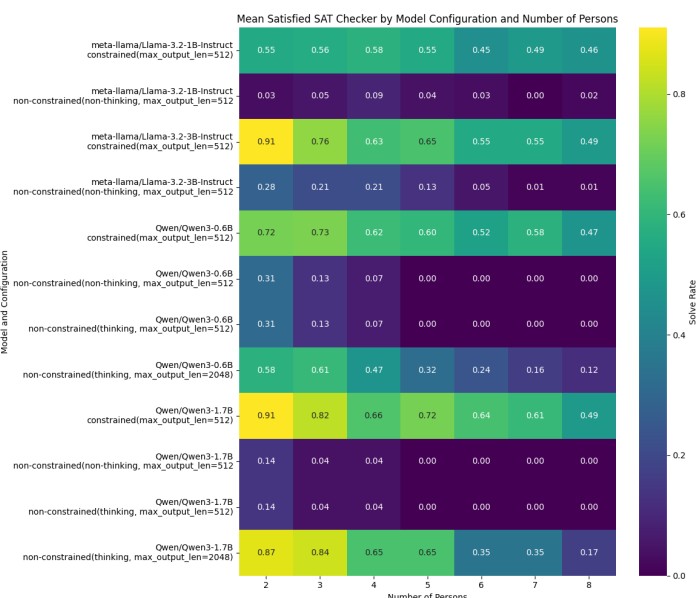

Figure 6: Knight and Knave test: Constrained models outperform unconstrained models across all puzzle sizes.

the "SELECT" clause. We also run experiments with the additional constraint of limiting identifier in the "SELECT" clause to the scope of all databases. Formally:

```
anchor: ID >> select_clause > select < from_clause << ID
def allow_containing_database(terminal_instance):
    return (□{}, ◊{name if name is an identifier of a database containing terminal_instance})

anchor: select_clause << ID
def correct_guess():
    return (□{}, ◊{name if is a table or column of any database })
```

For both constrained and unconstrained models, we remove tokens from the tokenizer that do not align with SQL grammar terminals, such as ".$Name$" and "$')$;". LLM performance is known to suffer when such tokens are not handled by constrainer (Beurer-Kellner et al., 2024). For the uncon-strained model, we add the target database schema to the model input, so the model has the same information on the target database that is available to the constrained model.

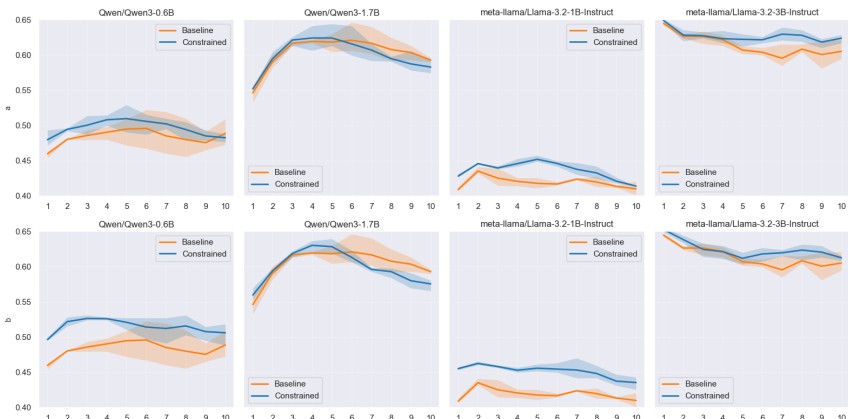

Figure 7: Text-to-SQL test: Execution accuracy of constrained vs non-constrained models over 3 runs and 10 epochs. Row $(a)$ uses selected columns in "SELECT" clause to limit the remaining identifiers to that of databases containing them. Row $(b)$ additionally constraints the "SELECT" clause to identifiers of all databases. Constrained decoding helps three of the four models.

The results are shown in fig. 7. "Qwen3-0.7B" , "Llama-3.2-1B-Instruct", and "Llama-3.2-3B-Instruct" benefit from the constrainer with either constraint configuration. However, "Qwen3-1.7B" does not benefit from constrainer.

## 6 CONCLUSION

We introduced a new context-sensitive grammar formalism, conditional deontics over terminals (CDoT), designed for use with constrained decoding, and a new $O(n^3)$ incremental CYK-like parser for the grammar. This constrained decoding framework is strong enough to implement the unit propagation algorithm, which we demonstrate on "Knight and Knave" logical puzzles, finding that our framework dramatically improves LLM accuracy on this dataset while simultaneously reducing the number of tokens the LLMs generate. We also apply our framework to constrain SQL table column names in a text-to-SQL task and find improved accuracy for most LLMs tested.

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

# A APPENDIX

## A.1 ADDITIONAL DETAILS OF KNIGHT AND KNAVE CONSTRAINER DESIGN

In the grammar for the Knight and Knave Constrainer Design, any special assignment sequence such as $x\_k\_condition$ is also acceptable as a plain *assignment_seq*. This undermines the new assignment we intend to force through our deontic rules. So we introduce the same delimiter at different priorities, defined as *semi_p* $\rightarrow$ SEMI_P NEWLINE in the grammar. Two terminals are needed to handle parser/LLM interfacing technicalities.

When a special assignment sequence is not complete, the following higher priority delimiter terminal will not be acceptable, consequently, the remaining part of special rule is not activated. On the other hand, when the conditions of a special assignment are met, the delimiter would be recognized as the higher priority version, and therefore deny the more general version of *assignment_seq*.

