# OpenReview forum: "Conditional Deontics over Terminals: A Mildly Context-Sensitive Formal Grammar for Constrained Decoding"
_ICLR.cc/2026/Conference — ICLR 2026 Conference Withdrawn Submission_

### Official Review · Reviewer_Sgbm · 2025-10-29

**Soundness:** 2
**Presentation:** 1
**Contribution:** 2
**Rating:** 2
**Confidence:** 3

**Summary:**

The paper presents a formalism for mildly context-sensitive grammar constrained decoding.  It addresses limitations of existing methods that require expensive backtracking or speculative lookahead by offering context-sensitive constraints with O(n³) time complexity and GPU acceleration. CDoT allows Python function invocation without requiring users to modify parser states. The paper shows improved performance on logical puzzles ("Knight and Knave") and text-to-SQL tasks while reducing computational costs compared to unconstrained decoding.

**Strengths:**

* The paper attempts to solve challenging context-sensitive grammar constrained-decoding problem. This is an important problem and any contribution in this direction can be impactful.

* The related work is reasonably comprehensive and covers most recent works constrained-semantic decoding. Although, I would suggest the authors to include more discussion context-free grammar decoding techniques.

**Weaknesses:**

## Presentation of Technical Contribution

> we design a new formal grammar for constrained-decoding

* What do you mean by design a new grammar? Do you mean a new class of grammars and a parser?


* The technical contribution of the paper is described in short section 4 with minimal detail. The exact technical contribution of the paper in comparison to the existing works on CYK parsing is unclear.


## Empirical results

* The evaluation considers two benchmarks. “Knight and Knave” and text-2-sql. I do not think both of these experiments sufficiently show the contribution improves over existing works. The comparison made in the paper is only against the unconstrained LLM and not against the existing SOTA baselines.

* Why were PICARD, Synchromesh or IterGen not considered for the experiment? What are practical advantages of proposed technique over the existing techniques?

* What class of practical constraints that cannot be handled by techniques that use backtracking or speculative lookahead such as PICARD, Synchromesh or IterGen?

* The experiments are limited to 4 small models

## Minor

> Text-to-SQL test: Execution accuracy of constrained vs non-constrained models over 3 runs and 10 epochs

What are epochs in this experiment?

**Questions:**

See above

---

> ### Author Response · Authors · 2025-12-02
>
> Hi! We will withdraw and undertake substantial revisions to this paper. Thank you very much for your careful and thoughtful reviews.

---

### Official Review · Reviewer_S4gW · 2025-10-30

**Soundness:** 2
**Presentation:** 3
**Contribution:** 2
**Rating:** 2
**Confidence:** 3

**Summary:**

This paper introduces Conditional Deontics over Terminals (CDoT), a new formal grammar extending context-free grammars with deontic operators  to capture conditional relationships between terminals during generation.
The authors build an incremental CYK-like parser capable of enforcing such constraints with O(n³) step complexity, and demonstrate applications to reasoning (“Knight and Knave”) and text-to-SQL constrained decoding.
The main claim is that CDoT enables a balance between tractability, context-sensitivity, and practical constrained decoding for large language models (LLMs).

The paper is clearly written and technically rigorous; however, it raises deep questions about the relevance and integration of symbolic grammar formalisms with modern neural generation. The work would benefit from reframing, empirical grounding, and conceptual modernization.

**Strengths:**

* The formalization of conditional deontic rules is novel and internally consistent. The distinction between necessary and permitted terminal conditions is both intuitive and flexible, providing an expressive way to encode hierarchical dependencies in symbolic grammars.
The examples (balanced meals, food–drink pairings) make the semantics accessible.
* The incremental parser is carefully designed, and the paper demonstrates solid knowledge of parsing theory (e.g., CYK, Valiant recognizer). The authors correctly maintain valid-prefix properties and discuss reachability states in detail.
The technical craftsmanship deserves credit even if the overall direction is debatable.
* The Knight and Knave setup is a creative use of logical reasoning to showcase constraint propagation, and the text-to-SQL experiment shows some awareness of practical downstream relevance. The integration of symbolic constraints with LLM inference is implemented competently.

**Weaknesses:**

* The biggest issue is not the internal coherence of the formalism but its ontological fit with LLMs.
Grammars operate over explicit derivations; LLMs operate over continuous, implicit distributions.
The CDoT framework effectively constrains token-level emissions post hoc, whereas current trends emphasize embedding-level control, latent constraint learning, or differentiable planning.

* Instead of presenting CDoT as a “new grammar formalism,” maybe more appropriate to reframe it as a constraint specification interface for LLM decoding, i.e., a structured programming API for safety or consistency enforcement.
That shift would make the work less about “reviving parsing” and more about interpretable constraint enforcement that complements modern decoding (e.g., beam search, logit masking, or self-consistency sampling).


* The “mildly context-sensitive” label feels borrowed from formal linguistics, but the paper never proves or empirically motivates that this level of context sensitivity is the right tradeoff for constrained decoding.
The O(n³) complexity result is mathematically elegant but not practically evaluated, and there’s no runtime data or ablation study demonstrating when the parser is faster, slower, or more expressive than existing systems like PICARD, IterGen, or XGrammar.


* In the Knight and Knave task, the CDoT parser performs reasoning itself (through unit propagation), which confounds whether the improvement comes from the model or from symbolic scaffolding.
This makes it hard to assess whether the grammar enhances LLM reasoning or replaces it.


* The paper risks sounding like it’s trying to “bring parsing back.” That narrative will face skepticism because the community has largely moved beyond discrete syntax control.
However, the idea of interpretable constraint management, especially for safety, correctness, or verifiability, is timely and highly relevant.

**Questions:**

* What is the precise motivation for using a formal grammar abstraction rather than a constraint graph or differentiable rule network?
Many modern decoding constraints are expressed at the token or semantic level (e.g., via logit masking, classifier guidance, or constraint satisfaction in embedding space). What unique representational advantage does the grammar formalism provide in this context?
* The paper claims that CDoT is mildly context-sensitive.
Can you formally characterize the class of languages it generates?
Does it strictly subsume CFGs but remain less powerful than full context-sensitive grammars?
A formal proof or even an intuitive inclusion diagram would clarify whether the “mildly” label is theoretically justified or purely descriptive.
* Why retain discrete syntactic machinery when LLMs already model syntax implicitly?
Could CDoT be reframed as an interpretable constraint layer that interacts with neural probability distributions, instead of a separate parsing engine?
* You mention that the parser involves “~20 modified incremental Valiant recognizers.”
Could you provide a clearer explanation or pseudocode outlining how these recognizers interact?
In particular, which components are responsible for deontic propagation, and which for structural validity?
It would help readers assess computational cost and reproducibility.
* The complexity claim (O(n³)) is interesting, but what is the constant factor in practice?
Do you have runtime comparisons against IterGen or XGrammar on equivalent decoding tasks, ideally using GPU-based implementations?
Such data would substantiate the claimed efficiency advantage.
* The LF_ALIVE, LF_REACHABLE↑, and LF_REACHABLE↓ states are key to your approach.
How do these attributes scale in memory usage as the sentence length or number of constraints grows?
Are there cases where they explode combinatorially, and how are these mitigated?

---

> ### Author Response · Authors · 2025-12-02
>
> Hi! We will withdraw and undertake substantial revisions to this paper. Thank you very much for your careful and thoughtful reviews.
>
> Our only concern is that parts of this review place discrete and continuous representations in an artificial opposition, rather than examining them through the lens of compositionality. Case in point, neural networks are implemented as compositions of discrete layers, a structure that underpins AutoGrad algorithms. These algorithms, in turn, make it possible to efficiently implement and evaluate a wide range of architectures, eventually enabling the development of Transformers. Recent constrained decoding systems exhibit a similar pattern of discrete-continuous composition: constraints are specified in a formal grammar, decomposed by a parser, before advanced techniques such as Monte Carlo sampling are employed.
>
> The suggestion to reframe is most helpful and insightful.
>
> We acknowledge the lack of formal treatment to the grammar, and will provide formal semantics and pseudo code in future revision.

---

### Official Review · Reviewer_rcnF · 2025-10-31

**Soundness:** 1
**Presentation:** 1
**Contribution:** 1
**Rating:** 2
**Confidence:** 5

**Summary:**

The paper presents a new context-sensitive grammar formalism, dubbed conditional deontics over terminals (CDoT). This type of grammars are intended to provide semantic control over the constrained LLM generation process. The authors also present a new incremental parser for the grammars based on Cocke–Younger–Kasami algorithm. They report cubic complexity with respect to the length of the parsed string. The experiments are performed on “Knight and Knave” logical puzzle and constrained decoding task of text-to-SQL.

**Strengths:**

* Presenting the concept of extending the context free grammars with additional constrains can be used for semantic constraining of LLM outputs.

**Weaknesses:**

* The presentation of the technique is incomplete.
* The time complexity bounds have not been  proved in the paper.
* The experiments are done on simple examples; a more comprehensive set of benchmarks is required
* The paper does not compare to the existing approaches.

**Questions:**

I appreciate that the paper brings up the techniques from context-sensitive grammars for constraining LLM generation. These techniques have often been overlooked for the complexity of specifying properties and time complexity.

However, the presentation of the proposed technique is incomplete, making it difficult to appreciate the contribution:
- The parser description is terse and does not give intuition about the correctness of the approach. It would be valuable to provide a “soundness” theorem, stating that the parser will accept the text that follows the user-specified rules.
- The interface of the parser to the LLM is not discussed.
- The translation from the domain specific language used for the examples to regular expressions is not discussed.
- The expressive power of deontic rules (i.e., what properties they can express) is not discussed
- The time bound of O(n^3) for a sequence length n is not proved in the paper.

The evaluation is performed on several simple examples. On the positive side, they give an indication that the approach can work, and produce better results than fully unconstrained generation on small open-source LLMs. However the evaluation does not compare to any existing work (even though some tools such as Itergen have been mentioned). The results for the test-to-SQL problem are presented as figures, but the paper provides only a brief description of the results and leaves out the conclusion of the benefits of the proposed approach.

My recommendation for the next revision of the paper is to extend the evaluation with additional benchmarks, compare the approach experimentally to existing tools, and present scaling of the parsing+checking for different sizes 'n'. Further, a closer comparison to the theoretical underpinning of existing constraint generation tools would highlight the unique aspects of your approach.

---

> ### Author Response · Authors · 2025-12-02
>
> Hi! We will withdraw and undertake substantial revisions to this paper. Thank you very much for your careful and thoughtful reviews.
>
> We acknowledge the lack of formal treatment to the grammar, and will provide formal semantics and pseudo code in future revision.

---

### Official Review · Reviewer_wDPe · 2025-10-31

**Soundness:** 3
**Presentation:** 3
**Contribution:** 3
**Rating:** 4
**Confidence:** 3

**Summary:**

To address challenges in constrained decoding for Large Language Models (LLMs)—such as the difficulty of implementing context-sensitive constraints, high computational costs, poor compatibility, and semantic risks of existing frameworks (e.g., IterGen)—this paper proposes CDoT (Conditional Deontics over Terminals), a lightweight context-sensitive formal grammar. Its incremental parser operates with a time complexity of \(O(n^3)\) per step and supports GPU acceleration. By leveraging deontic rules to constrain the necessary and permitted conditions of terminal instances, CDoT significantly improves the accuracy of LLMs and reduces token consumption in the "Knight and Knave" logical puzzle task, while also enhancing the accuracy of most tested LLMs in the Text-to-SQL task.

**Strengths:**

It accurately addresses the key challenges in constrained decoding for Large Language Models (LLMs): On one hand, it breaks through the bottleneck in implementing context-sensitive constraints. Through the design of "deontic rules + lightweight grammar", it avoids the complex logic of traditional solutions that rely on cross-reference across subtrees; on the other hand, it directly targets the shortcomings of existing frameworks (such as IterGen), solving problems including high computational costs, poor compatibility with speed optimization solutions (such as XGrammar), and the risk of over-constraints caused by overly permissive semantics. Its technical positioning is clear and practical.

**Weaknesses:**

- In the Text-to-SQL task, the Qwen3-1.7B model did not benefit from the CDoT constraint, and the paper did not clearly explain the core reason for this phenomenon, so its universality needs further verification.
- Although CDoT simplifies constraint definition through the "anchor + deontic function" design, for complex tasks (such as multi-turn logical reasoning and nested SQL generation), the design of deontic rules may require refined adjustments (e.g., conditional association of multiple antecedent terminals, writing of regular expressions for complex necessary conditions), which means users need to have a certain level of proficiency in grammar design and logical modeling.

**Questions:**

It is necessary to compare some of the latest related works, such as:

Sun, Xintong, et al. "Earley-Driven Dynamic Pruning for Efficient Structured Decoding." Forty-second International Conference on Machine Learning.

Regarding constraints on nested structures (e.g., the matching between subquery columns and outer query tables in SQL), can the "left-foot state tracking" mechanism of CDoT effectively cover cross-hierarchical terminal associations?

---

> ### Author Response · Authors · 2025-12-02
>
> Hi! We will withdraw and undertake substantial revisions to this paper. Thank you very much for your careful and thoughtful reviews.
>
> We were not previously aware of the Earley-based system you mentioned, and we will ensure that it is properly cited and discussed in a revised version.
>
> We also acknowledge that the behavior of CDoT in more complex scenarios is currently underspecified. We will address this by providing formal semantics for CDoT in a future revision.

---

### Note · Authors · 2025-12-02

I have read and agree with the venue's withdrawal policy on behalf of myself and my co-authors.